# MIRAGE: A Benchmark for Multi-hop Interleaved Reasoning and Retrieval-Grounded Evidence

## ABSTRACT

Large Language Models (LLMs) are increasingly expected to function as autonomous agents that can solve complex, multi-step problems. This requires a dynamic interplay of retrieving external knowledge and reasoning upon the gathered evidence. However, current benchmarks typically evaluate these capabilities in isolation, using static, single-hop retrieval tasks or closed-book reasoning tests, failing to capture the critical skill of interleaving these processes. To bridge this evaluation gap, we introduce MIRAGE, a new, challenging benchmark for **M**ulti-hop **I**nterleaved **R**easoning **a**nd **G**rounded **E**vidence. Constructed from real-world conversational data, MIRAGE comprises 579 complex tasks that require agents to autonomously formulate a sequence of sub-queries, retrieve information over multiple turns, and synthesize evidence in the presence of curated adversarial distractor documents. Our comprehensive evaluation of state-of-the-art systems reveals a stark architectural divide: dynamic, interleaved RAG agents dramatically outperform conventional single-turn RAG pipelines. Despite this, even the most advanced reasoning-intensive retrieval frameworks struggle, achieving a final task success rate of less than 30%. Our analysis pinpoints critical failure modes in both agentic planning and evidence synthesis, highlighting key challenges for future research. We release MIRAGE as a public resource to catalyze the development of more robust and capable reasoning agents.

## 1 INTRODUCTION

Real-world problem-solving is rarely a single-shot task. Consider a financial analyst investigating a company's performance decline: their process is not a single query but an iterative loop of forming a hypothesis (*e.g.*, "Is the decline due to supply chain issues?"), retrieving reports, reasoning over the retrieved data to identify gaps, and then formulating new sub-queries to seek further evidence (*e.g.*, "Find reports on their key suppliers' performance."). This dynamic **interleaving** of retrieval and reasoning is fundamental to complex cognition.

While the paradigm of Retrieval-Augmented Generation (RAG) (Lewis et al., 2020a) has become the standard for grounding LLMs in external knowledge, it is often applied in a static, one-off manner: retrieve, then synthesize an answer. This falls short of the ambitious vision pursued by deep research programs in projects like DeepResearch (2025), which aim to build agents capable of tackling complex, multi-step research and analysis tasks autonomously. Although powerful reasoning-focused models like DeepSeek-R1 (Guo et al., 2025), Gemini-Pro (Comanici et al., 2025), and OpenAI's GPT-5 (Jaech et al., 2024) provide the engine for this vision, the methods for evaluating them remain disconnected from this iterative, agentic process. Existing benchmarks fall short: (1) **Reasoning benchmarks** like GSM8k (Cobbe et al., 2021) are "closed-book," ignoring retrieval entirely. (2) **Multi-hop QA benchmarks** like HotpotQA (Yang et al., 2018) evaluate static, pre-defined reasoning chains. (3) **Reasoning-intensive retrieval benchmarks** like BRIGHT (Su et al., 2025) focus on the quality of a single, complex retrieval step, not the sequential decision-making process. This disconnection raises a fundamental research question: *how can we rigorously measure and drive progress on the essential **agentic skill** of **dynamically interleaving reasoning and retrieval** over **multiple turns**?*

Table 1: Comparison of MIRAGE with existing benchmarks at the intersection of retrieval and reasoning. MIRAGE is the first to holistically evaluate the interleaved process of agentic decision-making, multi-hop synthesis, and resilience to adversarial distractors.

| Benchmark | No. QAs | Multi-hop | Reasoning Intensive | Adversarial Distractors | Agentic Decision-Making | Interleaved Process |
|---|---|---|---|---|---|---|
| *Primarily Retrieval-Focused Benchmarks* | | | | | | |
| BEIR (Thakur et al., 2021) | ∼150k | ✗ | ✗ | ✗ | ✗ | ✗ |
| BRIGHT (Su et al., 2025) | 1.4k | ✗ | ✓ | ✗ | ✗ | ✗ |
| RAR-b (Xiao et al., 2024) | 5k | ✗ | ✓ | ✗ | ✗ | ✗ |
| LexRAG (Li et al., 2025a) | 1013 | ✗ | ✗ | ✓ | ✗ | ✗ |
| *Primarily Reasoning-Focused Benchmarks (with provided evidence)* | | | | | | |
| HotpotQA (Yang et al., 2018) | 113k | ✓ | ✗ | ✗ | ✗ | ✗ |
| 2WikiMultihopQA (Ho et al., 2020a) | 192K | ✓ | ✗ | ✗ | ✗ | ✗ |
| *Interleaved / Conversational Benchmarks* | | | | | | |
| MTRAG (Katsis et al., 2025) | 110 | ✗ | ✗ | ✗ | ✓ | ✗ |
| MultiHop RAG (Tang & Yang, 2024) | 2556 | ✓ | ✗ | ✗ | ✓ | ✗ |
| **MIRAGE (Ours)** | **579** | ✓ | ✓ | ✓ | ✓ | ✓ |

To address this critical evaluation gap, we introduce MIRAGE (**M**ulti-hop **I**nterleaved **R**easoning **a**nd **G**rounded **E**vidence), a new benchmark designed to rigorously assess the ability of LLMs to perform complex, multi-turn tasks that necessitate a fluid interplay between multi-hop information retrieval and evidence-grounded reasoning. MIRAGE is constructed from real-world scenarios across four challenging domains: Legal, Finance, Technology, Academia, featuring questions in Legal Case Analysis, Legal Article Matching, Financial consulting, Scientific Literature Review, Software Engineering and Development, Hardware Debugging, *etc*. It comprises 579 complex question-answering scenarios, each requiring multiple interaction turns. Crucially, each scenario is grounded in a corpus containing not only the necessary evidence documents but also a set of curated, high-similarity **distractor documents**. These distractors are thematically related but irrelevant to the question, specifically designed to challenge the model's precision and penalize superficial keyword matching.

We conduct a comprehensive evaluation of state-of-the-art models and retrieval strategies on MIRAGE. Our findings reveal a significant performance deficit in current systems. Even sophisticated agentic frameworks built on powerful LLMs struggle to navigate the challenges posed by MIRAGE, with the best-performing approach achieving a final task success rate of only 61.5%. Our analysis reveals two primary failure modes: **premature reasoning**, where models attempt to synthesize an answer before all necessary evidence has been collected, and **distractor vulnerability**, where the reasoning process is derailed by plausible but incorrect information from distractor documents.

## 2 RELATED WORK

Our work is positioned at the intersection of two major lines of research in language model evaluation: benchmarks for information retrieval and benchmarks for multi-step reasoning. We argue that while both fields have seen significant advances, they have largely evolved in isolation, leaving the critical, intertwined process of dynamic retrieval and reasoning under-evaluated.

**The Evolution of Retrieval Benchmarks** The evaluation of information retrieval has progressed from testing simple semantic similarity to assessing complex, knowledge-intensive tasks. Foundational benchmarks like BEIR (Thakur et al., 2021) and MTEB (Muennighoff et al., 2022) established comprehensive leaderboards for single-hop retrieval across diverse domains. Responding to the growing context capabilities of LLMs, recent work has expanded into long-context retrieval (Xu et al., 2023; Li et al., 2023). A more recent and relevant research thrust focuses on "reasoning-intensive" retrieval. Benchmarks like BRIGHT (Su et al., 2025) and RAR-b (Xiao et al., 2024) present queries where lexical or semantic overlap is insufficient, requiring a model to perform inferential steps to identify the relevant documents. While these represent a significant step forward, their focus remains on the quality of a **single retrieval turn**. They evaluate a model's ability to find the right evidence, but not its ability to use that evidence to decide what to look for next in an iterative process.

**The Evolution of Grounded Reasoning Benchmarks** Concurrent to advances in retrieval, reasoning benchmarks have moved from "closed-book" assessments to "open-book," evidence-grounded formats. Early benchmarks such as GSM8K (Cobbe et al., 2021) and HumanEval (Chen et al., 2021) test a model's internal deductive capabilities on math or coding in a vacuum. To better reflect real-world tasks, multi-hop QA benchmarks like HotpotQA (Yang et al., 2018) and 2Wiki-MultiHopQA (Ho et al., 2020b) were introduced. These were pivotal in requiring models to synthesize answers from multiple provided source documents. However, they operate on the assumption of a perfect oracle; the necessary documents are pre-selected and provided to the model. They test the ability to reason over given evidence, but not the critical skill of **autonomously seeking that evidence** in the first place.

**The Gap: Evaluating Interleaved Agentic Processes** The limitations of these two research tracks highlight a critical gap: the evaluation of the complete, interleaved agentic process. A nascent category of benchmarks is beginning to explore this. Conversational RAG datasets such as MTRAG (Katsis et al., 2025) evaluate retrieval over multiple turns but often prioritize dialogue coherence over solving complex, multi-hop reasoning problems. Broader agentic benchmarks (Wei et al., 2025; Mialon et al., 2023) test tool use in complex environments but are often not focused on the specific, tight loop of text-based retrieval and reasoning (Li et al., 2025b). As summarized in Table 1, **MIRAGE** is the first benchmark designed specifically to fill this void. It uniquely combines the challenges of reasoning-intensive retrieval with the necessity of multi-step, grounded reasoning, all within a dynamic framework that demands agentic decision-making at each turn and resilience against adversarial distractors.

## 3 THE **MIRAGE** BENCHMARK

Constructing a benchmark that faithfully evaluates interleaved reasoning and retrieval requires a departure from traditional data collection methods. In this section, we first provide a formal definition of the task we aim to evaluate (§3.1). We then detail our three-stage data curation pipeline designed to transform fragmented, real-world conversations into complex, standalone reasoning tasks (§3.2). Finally, we present the statistics and composition of the resulting benchmark (§3.3).

### 3.1 TASK DEFINITION

The core task in **MIRAGE** is to generate a correct and well-supported final answer $A$ for a complex, high-level question $Q$. This cannot be achieved in a single step. Instead, an agent must engage in a multi-turn process of reasoning and retrieval. Formally, at each turn $i$, the agent is given the initial question $Q$ and the history of its previous actions $H_i = \{(q_1, D_1, a_1), \ldots, (q_{i-1}, D_{i-1}, a_{i-1})\}$. The agent must then make a decision:

1. **CONTINUE**: If the problem is not yet solved, the agent must reason upon $Q$ and $H_i$ to formulate a new, targeted sub-query $q_i$. This sub-query is used to retrieve a set of documents $D_i$ from a large corpus $\mathcal{C}$. The agent then synthesizes the information in $D_i$ to produce an intermediate answer $a_i$.
2. **HALT & ANSWER**: If the agent determines it has gathered sufficient evidence, it synthesizes the information from the entire history $H_i$ to generate the final answer $A$.

A successful trajectory requires proficiency in both **iterative planning** (formulating the correct sequence of sub-queries) and **evidence-grounded reasoning** (accurately synthesizing retrieved information at each step). The corpus $\mathcal{C}$ for each question contains a small set of positive documents $D^+$ and a much larger set of adversarial distractor documents $D^-$, making precise retrieval a significant challenge.

### 3.2 DATA CURATION PIPELINE

To create instances that embody this task, we developed a three-stage pipeline to systematically transform real-world conversational data into high-quality benchmark tasks, as depicted in Figure 1.

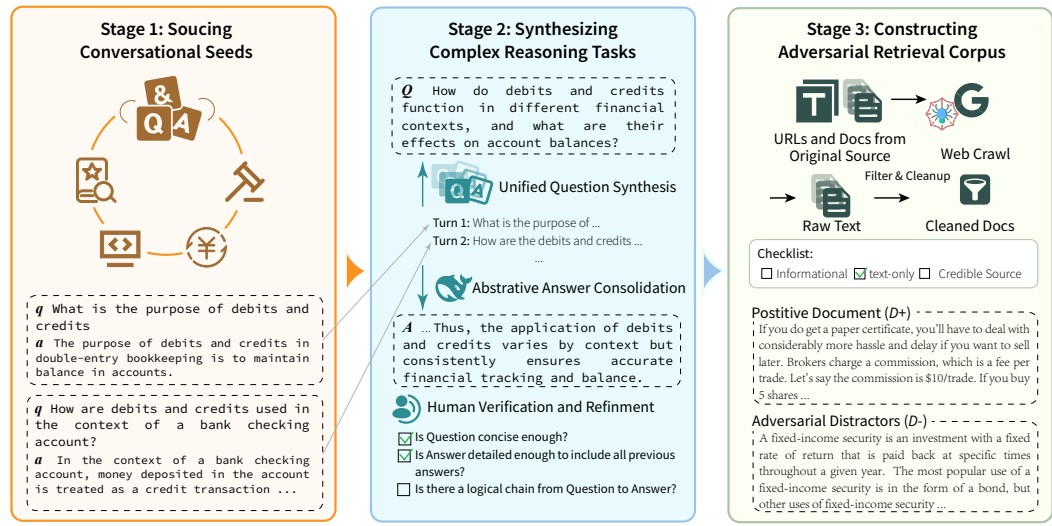

Figure 1: The three-stage data curation pipeline for MIRAGE. We begin with raw **Conversational Seeds** from real-world sources (§3.2.1), **Synthesize** them into complex, coherent reasoning tasks using an LLM-in-the-loop process (§3.2.2), and finally construct a bespoke **Adversarial Retrieval Corpus** for each task (§3.2.3).

### 3.2.1 STAGE 1: SOURCING CONVERSATIONAL SEEDS

**Data Source** We began by sourcing "conversational seeds"—naturally occurring multi-turn discussions that contain an implicit, multi-step reasoning process. Our sources were two fold:

- **Stack Exchange Forums:**[1] We targeted knowledge-intensive forums such as Law, Finance, Computer Science, and Academia, leveraging the public data dump from the Internet Archive.
- **Existing Datasets:** To broaden our domain coverage, we extracted relevant multi-turn fragments from existing high-quality datasets, including Doris-Mae (Wang et al., 2023), LexRAG (Li et al., 2025a), and MTRAG (Katsis et al., 2025).

**Preprocessing** From this initial pool, we applied a rigorous filtering and validation protocol to isolate only the highest quality seeds suitable for our benchmark. Each candidate conversation was required to meet three key criteria:

1. **Iterative Structure:** The conversation had to contain more than two turns to ensure a genuine iterative process. For Stack Exchange data, we enforced a strict validation rule: the same username had to appear in multiple turns of a single thread, confirming a sustained, context-linked interaction rather than a series of disconnected, single-turn responses.
2. **Evidence Grounding:** The thread was required to contain explicit URL references to external documents. We then manually verified these references to ensure they were active and credible, filtering out any seeds that relied on broken links or low-quality sources.
3. **Data Quality:** Finally, each selected seed underwent a cleaning process to remove undesirable noise (*e.g.*, excessive special characters, formatting artifacts) and standardize the text to facilitate downstream processing.

### 3.2.2 STAGE 2: SYNTHESIZING COMPLEX REASONING TASKS

The raw conversational seeds are fragmented by nature. The core of our contribution lies in transforming these fragments into coherent, standalone reasoning tasks through a combination of advanced LLM synthesis and expert human oversight.

1. **Unified Question Synthesis.** For each conversational seed, we aggregated all user questions and prompts into a single, comprehensive query $Q$. We used DeepSeek-R1 (Guo et al., 2025)

---

[1] https://archive.org/details/stackexchange

to synthesize these inputs into a high-level question that captures the user's ultimate information need, abstracting away the conversational back-and-forth. This consolidation methodology eliminates redundant queries, optimizing the information retrieval process. It produces clearer, more focused search objectives that enhance downstream analysis effectiveness. The resulting streamlined query structure significantly improves the efficiency of subsequent reasoning while maintaining the original dialogue's informational completeness.

2. **Abstractive Answer Consolidation.** Similarly, we consolidated all the intermediate answers and information from the thread into a single, definitive final answer $A$. Each response within the multi-turn conversation is analyzed to extract relevant information while eliminating redundant, conflicting, or inconsequential content. This process is *abstractive*: the final answer is not merely a summary but a complete synthesis that resolves contradictions and presents the information logically, grounded in the evidence from the intermediate steps. This abstraction enables the final answer to effectively address the overarching question synthesized in the previous step, even if the phrasing or structure diverges from earlier outputs. In this way, the consolidation step functions as both an aggregator and a reasoner, aligning fragmented evidence into a unified response that is both *contextually grounded and logically consistent*. This synthesized response preserves the original contextual integrity while presenting a more concise and informative summary.

3. **Human Verification and Refinement.** Each synthesized $(Q, A)$ pair, along with its intermediate reasoning trace (the original sub-questions and their referenced documents), was reviewed by human experts. Annotators verified factual correctness, logical coherence between steps, and that the final answer $A$ was fully supported by the evidence in the trace. Pairs that were ambiguous or factually inconsistent were either corrected or discarded. Further details on the annotation guidelines are available in Appendix C.

### 3.2.3 STAGE 3: CONSTRUCTING THE ADVERSARIAL RETRIEVAL CORPUS

A key challenge in **MIRAGE** is navigating a noisy information environment. For each task, we constructed a specific retrieval corpus $\mathcal{C}$ by systematically curating positive documents and a much larger set of targeted, adversarial distractors.

**Positive Documents** ($D^+$) These are the documents explicitly and correctly referenced in the original conversational seed, as verified during Stage 2. Given their predominantly web-based nature, each positive document was meticulously archived to prevent link rot and indexed to ensure consistent and accurate retrieval during evaluation.

**Adversarial Distractors** ($D^-$) To rigorously test model precision and move beyond simple semantic matching, we designed a two-pronged strategy for creating challenging negative documents:

1. *In-Domain Distractors:* We included positive documents from other tasks within the same domain (*e.g.*, other legal cases for a legal query). This approach leverages documents that are thematically relevant and share domain-specific jargon, acting as highly plausible yet incorrect evidence that tests a model's ability to discern fine-grained relevance.
2. *Topical Distractors:* To generate distractors that are semantically similar to the ground truth, we employed a targeted web search strategy. For each positive document, we used an LLM to generate a concise topic summary, which was then used as a query in Google Search. We then extracted the top-k (e.g. k=10) search results that were distinct from the original positive document.

This dual-pronged strategy ensures that the negative set is not merely random noise but a high-quality collection of plausible, contextually relevant distractors, forcing models to reason deeply about the information they retrieve.

### 3.3 BENCHMARK STATISTICS

The final benchmark consists of 579 complex question-answering tasks distributed across four primary domains. Each task instance includes: (1) a high-level, synthesized question $Q$; (2) the gold sequence of intermediate sub-questions $\{q_i\}$; (3) the set of positive documents $D^+$ required for the solution; (4) the final, consolidated answer $A$; and (5) a dedicated retrieval corpus $\mathcal{C}$ containing $D^+$ and a large set of adversarial distractors $D^-$. The detailed statistics are presented in Table 2.

Table 2: **Statistics of MIRAGE across its four domains.** '$Q$' denotes the number of complex tasks. '$D+$' is the total number of unique positive documents. '$\mathcal{C}$' is the total corpus size per task (positive + distractors). 'Avg. Hops' is the average number of retrieval turns per task.

| Domain | # Tasks (Q) | # Pos. Docs (D+) | Avg. Corpus Size ($\mathcal{C}$) | Avg. Hops | Primary Sources | Examples |
|---|---|---|---|---|---|---|
| Finance | 69 | 260 | ~1,600 | 3.8 | StackExchange, MTRAG | F |
| Legal | 216 | 808 | ~1,450 | 3.4 | StackExchange, LexRAG | G |
| Technology | 184 | 612 | ~1,600 | 5.2 | StackExchange, MTRAG | E |
| Academia | 110 | 520 | ~1,000 | 3.6 | StackExchange, Doris-Mae | D |
| **Total** | **579** | ~2,200 | ~5,450 | 4 | – | |

## 4 EXPERIMENTS

In this section, we conduct a series of experiments to establish the difficulty of **MIRAGE** and diagnose the failure modes of current state-of-the-art systems. Our investigation follows a logical progression: we first demonstrate that retrieval is essential by evaluating powerful LLMs in a closed-book setting (§4.1). We then show that standard, single-shot Retrieval-Augmented Generation (RAG) is insufficient (§4.2). Finally, we benchmark advanced, multi-step agentic frameworks to reveal their core challenges in planning and synthesis (§4.3).

### 4.1 BASELINE: RETRIEVAL IS NECESSARY

To confirm that the tasks in **MIRAGE** cannot be solved using parametric knowledge alone, we first evaluate a range of powerful LLMs in a direct, closed-book generation setting.

**Setup.** We evaluate leading open-source and proprietary models, including Qwen-3-14B, Llama-3-8B, DeepSeek-R1, and API-based models like GPT-o3 and Claude 3.5. We test models with and without chain-of-thought (CoT) prompting to elicit their maximum reasoning capabilities. We use a suite of metrics to assess the quality of the response, including factual correctness (Fact-Score), exact match (Str-EM), and semantic similarity (BERTScore).

**Results and Analysis.** As shown in Table 3, all models perform poorly, with the best model, GPT-o3 (with CoT), achieving a Fact-Score of only 50.7. This confirms that parametric knowledge is insufficient and that **retrieval is a prerequisite for success on MIRAGE**.

An important insight comes from the uniformly low BERTScore across all models. The ground-truth answers in **MIRAGE** are abstractive summaries synthesized from multiple documents. Closed-book models, even when they hallucinate partially correct facts, fail to replicate this complex, evidence-grounded structure, resulting in low semantic similarity. This validates the design of our benchmark's abstractive answer consolidation process.

Table 3: Direct Generation Results

| Model | FactS. | Str-EM | Rouge | BERTS. |
|---|---|---|---|---|
| Non-thinking | | | | |
| DeepSeek V3 | 49.6 | 30.5 | 17.4 | 13.7 |
| GPT-4.1 | 51.3 | 33.1 | 19.6 | 21.9 |
| Calude 3.5 | 49.8 | 31.2 | 18.2 | 21.5 |
| Qwen-2.5-3B | 10.6 | 10.5 | 9.5 | 8.3 |
| Llama-3-3B | 10.4 | 11.2 | 9.4 | 8.2 |
| Qwen-2.5-7B | 32.5 | 22.3 | 13.7 | 9.5 |
| Qwen-3-14B | 41.5 | 21.5 | 15.2 | 12.5 |
| With Thinking | | | | |
| DeepSeek R1 | 50.1 | 30.4 | 18.2 | 18.2 |
| Qwen-3-14B | 42.6 | 21.7 | 16.9 | 13.6 |
| OpenAI o3 | 50.7 | 22.4 | 20.2 | 10.0 |

### 4.2 STANDARD RAG IS INSUFFICIENT FOR MULTI-HOP COMPLEXITY

Next, we investigate whether a standard, single-shot "retrieve-then-read" RAG pipeline can solve the tasks. This represents the most common approach to knowledge-grounded generation today.

**Setup.** We pair a comprehensive set of retriever models with a fixed generator LLM (Qwen-3-8B). The retriever's task is to fetch the top-k documents using the high-level question $Q$ as the sole query.

The generator then synthesizes an answer from these retrieved documents. We evaluate retriever performance using nDCG@10 and Recall, and end-to-end task performance using Fact-Score.

**Results and Analysis.** The results in Table 4 reveal two critical bottlenecks. First, all retrievers struggle significantly, with the best model, ReasonIR (Shao et al., 2025b), achieving an nDCG@10 of only 22.1. This demonstrates that the initial high-level question in **MIRAGE** is intentionally abstract and does not contain the specific keywords needed to retrieve all required evidence in a single shot. Second, this poor retrieval performance translates directly to poor end-to-end results. The best RAG pipeline achieves a Fact-Score of only 51.2, a marginal improvement over the closed-book baseline, and this high score is also attributable to the model's rich inherent knowledge instead of the rich external knowledge we expect the model to use. This confirms that **a simple retrieve-then-read paradigm is insufficient; a dynamic, multi-hop process is required.**

Table 4: Performance comparison between different retrievers. The Reasoner we used is Qwen-2.5-7B without the thinking capability.

| Retriever | nDCG@10 | Precision | Recall |
|---|---|---|---|
| *Sparse Model* | | | |
| BM25 | 12.5 | 10.1 | 9.5 |
| *Open-sourced Models (Small)* | | | |
| BGE-m3 | 14.5 | 11.8 | 10.1 |
| E5-base-v2 | 14.3 | 12.1 | 11.7 |
| SBERT | 15.2 | 13.2 | 12.5 |
| *Open-sourced Models (Large)* | | | |
| Inst-XL | 14.8 | 13.5 | 13.8 |
| Qwen | 15.1 | 14.5 | 14.0 |
| GritLM | 15.1 | 14.8 | 14.1 |
| ReasonIR | 22.1 | 21.1 | 21.5 |
| *Proprietary models* | | | |
| OpenAI | 20.6 | 19.8 | 19.6 |
| Google | 19.6 | 19.5 | 19.9 |

### 4.3 REASONING-BASED RETRIEVAL STRUGGLE WITH PLANNING AND SYNTHESIS

Having established that multi-step interaction is necessary, we evaluate a range of SOTA reasoning-based retrieval frameworks, which has demonstrated their successful ranks on BRIGHT (Su et al., 2025) benchmark.

**Setup.** We evaluate several advanced reasoning-based retrieval pipelines, including XRR-2 (Jataware, 2023), RaDeR (Das et al., 2025), ReasonRank (Liu et al., 2025) and TongSearch (Qin et al., 2025), which combine retrieval, reranking, and generation in an iterative loop. Each model starts with the question $Q$ and must autonomously generate sub-queries, retrieve evidence, and synthesize findings until it decides to produce a final answer. Our primary metric is **End-to-End Success Rate**, which measures the percentage of tasks where the final answer is factually correct and fully supported by the retrieved evidence. We also conduct a detailed error analysis to diagnose failure modes.

Table 5: Performance comparison between different retriever pipelines with rerankings.

| Retriever | F1 | Precision | Recall |
|---|---|---|---|
| XRR-2 | 37.5 | 36.5 | 37.4 |
| RaDaR | 35.6 | 35.4 | 35.8 |
| ReasonRank | 33.4 | 33.2 | 34.6 |
| TongSearch | 31.1 | 30.9 | 31.4 |

**Results and Analysis.** As summarized in Table 5, even SOTA retrieval frameworks find **MIRAGE** extremely challenging. The best performing system, XRR-2, achieves a F1 score of only 37.5. While this is a significant improvement over standard RAG, it underscores the deep challenges that remain. Our error analysis reveals two primary failure modes:

- **Planning Failures** ($\sim 75\%$ **of errors):** Agents frequently struggle to formulate effective sub-queries. We observe instances of agents getting stuck in retrieval loops, asking irrelevant questions, or halting prematurely before all necessary evidence has been gathered (premature grounding).
- **Synthesis Failures** ($\sim 25\%$ **of errors):** In many cases, the agent successfully retrieves the correct documents but fails to reason over them. A common cause is being misled by our **adversarial distractors**, where an agent incorrectly privileges a plausible but irrelevant document over the correct one.

Table 6: **Main Results on Various RAG Architectures.** FS (Fact-Score) and Str-EM (String Exact Match) measure end-to-end generation quality. BS (BERTScore), nDCG@10, Pre (Precision), and Rec (Recall) are also reported. All scores are averaged across all domains. Best in category is bolded. Overall best academic system is highlighted. We used the same retriever ReasonIR for all the scheme in single-turn RAG category.

| Category | Model | Reasoner Metrics | | | Retriever Metrics | | |
|---|---|---|---|---|---|---|---|
| | | FS | Str-EM | BS | nDCG@10 | Pre | Rec |
| Single-Turn RAG | Qwen-3 | 41.2 | 34.3 | 19.9 | 22.1 | 21.1 | 21.5 |
| | Qwen-3+thinking | 42.3 | 35.8 | 20.4 | 22.1 | 21.1 | 21.5 |
| | OpenAI o3 | 51.0 | 42.3 | 22.7 | 22.1 | 21.1 | 21.5 |
| | OpenAI o3+thinking | 51.1 | 49.1 | **23.5** | 22.1 | 21.1 | 21.5 |
| | DeepSeek V3 | 51.2 | 41.2 | 18.3 | 22.1 | 21.1 | 21.5 |
| | DeepSeek R1 | 50.7 | 41.0 | 19.6 | 22.1 | 21.1 | 21.5 |
| Multi-Hop RAG | Self-RAG (Asai et al., 2023) | 60.3 | 41.3 | **28.1** | **39.8** | **38.6** | **39.4** |
| | SeakR (Yao et al., 2024) | 59.1 | 41.0 | 27.9 | 36.7 | 36.5 | 36.8 |
| | Multi-hop RAG | 41.5 | 39.1 | 20.3 | 15.5 | 16.3 | 14.2 |
| Interleaving RAG | Search-R1 (Jin et al., 2025) | 61.5 | 40.5 | **38.3** | **42.3** | **41.3** | **43.6** |
| | R-Search (Zhao et al., 2025) | 60.4 | 39.8 | 35.6 | 41.8 | 40.7 | 41.9 |
| | Re-Call (Chen et al., 2025) | 60.2 | 37.6 | 33.7 | 40.9 | 40.5 | 40.6 |
| Deep Search | OpenAI Assistant API* | 75.2 | 45.3 | **40.1** | - | - | - |
| | Jina AI* | 71.2 | 34.5 | 39.4 | - | - | - |
| | Gemini Lirve* | 73.1 | 40.1 | 39.4 | - | - | - |
| | Grok* | 76.2 | 40.2 | 39.4 | - | - | - |

\* Deep Search products were evaluated via their public APIs, which include web search; retrieval metrics are not applicable.

These findings highlight that **MIRAGE** is an effective tool for stressing the critical planning and synthesis capabilities of advanced agents, revealing key weaknesses that must be addressed by future research.

## 4.4 ANALYSIS: FROM STATIC RETRIEVAL TO DYNAMIC RETRIEVAL AGENTS

To understand the key drivers of performance on **MIRAGE**, we conduct a deeper analysis centered on the architectural choices of different RAG systems (Lewis et al., 2020b). We categorize existing systems into a four-part typology to systematically dissect their strengths and weaknesses, *i.e.*, **i.** vanilla single-turn RAG; **ii.** multi-hop RAG with a pre-defined multi-retrieval process; **iii.** interleaved RAG where the LLM dynamically controls when and what to retrieve; and **iv.** commercial deep search products that leverage web search, representing a practical upper-bound.

The results in Table 6 reveal a stark architectural divide. Further implementation details for all evaluated systems can be found in Appendix B. The choice of RAG architecture is the single most important factor for success on **MIRAGE**, with dynamic, interleaved systems dramatically outperforming their static, single-turn counterparts. We picked six RAG systems to evaluate as our main results. We followed the instructions from the original work, and we used the pre-trained reasoners and retrievers from the original work.

- **Single-Turn RAG Establishes a Low Baseline.** Conventional RAG pipelines, which use the high-level question for a single retrieval step, perform poorly. Even when paired with powerful generators like GPT-o3, the Fact-Score remains low. This is primarily due to the failure of the initial retrieval step (nDCG@10 of 22.1), confirming that the complex questions in **MIRAGE** cannot be resolved with a single information-seeking turn.
- **Multi-Hop RAG Shows Incremental Gains.** Structured multi-hop systems like Self-RAG show a significant improvement in retrieval metrics (*e.g.*, 39.8 nDCG@10 for Self-RAG), as they can gather more evidence across fixed steps. However, their generation scores, while better, do not see a commensurate leap, suggesting that their rigid, pre-programmed retrieval strategies are not as effective as fully dynamic planning.

- **Interleaving RAG Demonstrates SOTA Performance.** The highest-performing academic methods fall into this category. Systems like Search-R1 (Jin et al., 2025) achieve the best balance of retrieval and generation, with a Fact-Score of 61.5 and an nDCG@10 of 42.3. This highlights the superiority of an agentic approach where the LLM has full control to adapt its retrieval strategy based on the information it discovers at each turn.
- **Deep Search Products Validate the Paradigm.** Black-box commercial systems from providers like OpenAI and Jina AI, which embody the interleaved web search paradigm, set the performance ceiling. Their high generation scores underscore the real-world effectiveness of dynamic, agentic information-seeking, validating the core principles tested by MIRAGE.

**Within a given architecture, does explicit reasoning help?** The main results table allows us to perform an ablation study on the effect of chain-of-thought (CoT) reasoning. In the Single-Turn RAG category, enabling a "thinking" step for models like Qwen3 and GPT-o3 provides a consistent but modest performance boost (*e.g.*, +0.8 BertScore for GPT-o3). This indicates that while the overall agentic architecture is the primary driver of success, improving the quality of reasoning at each individual synthesis step still yields valuable marginal gains.

### 4.4.1 CASE STUDY: HUMAN VS. MACHINE REASONING PATHS

To understand *why* interleaved agents perform better, we qualitatively analyze the reasoning paths they generate.

**Divergence in Sub-Query Strategy.** We compare the gold, human-authored sub-queries with those generated by the top-performing agent, Search-R1. As illustrated in Appendix H, we observe a distinct pattern: human reasoning paths often involve making larger conceptual leaps and asking more abstract sub-questions. In contrast, the LLM agent tends to follow a more conservative, incremental path, breaking the problem down into smaller, more literal steps. While effective, this suggests that current agents still lack the capacity for the kind of abstract task decomposition that is intuitive to humans. The detailed comparison between human sub-query chain and model sub-query chain can be found in Appendix H.

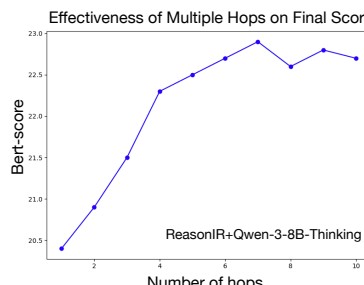

Figure 2: F1 score of RAG method on Finance domain data related to different number of hops.

**Performance vs. Reasoning Hops.** We further investigate the impact of *multi-turn reasoning* on a model's ability to generate accurate answers. Specifically, we examine how varying the number of reasoning turns affects the performance of a RAG system within the technology domain, where complex, multi-step reasoning is frequently required. In Figure 2, we demonstrate that agent performance within the technology domain correlates positively with the number of reasoning "hops" or turns. However, the performance gains begin to plateau after approximately 5-6 turns. This suggests that while agents benefit from the iterative process, they may not yet be efficient in their reasoning, sometimes taking more steps than necessary or failing to effectively integrate information from a long chain of evidence.

## 5 CONCLUSION

We introduce MIRAGE, a novel benchmark designed to address a critical gap in LLM evaluation: the ability to dynamically interleave multi-hop retrieval and evidence-grounded reasoning. By constructing complex tasks from real-world scenarios and incorporating a challenging suite of adversarial distractors, we create a testbed that pushes the boundaries of current agentic retrieval capabilities. Our extensive experiments revealed that the next frontier of progress in this domain may lie not in incremental improvements to individual retriever or generator modules, but in enhancing the holistic, agentic capabilities of the models themselves. We hope that MIRAGE will enable more rigorous evaluation and inspire the development of the next generation of AI agents capable of tackling the complex, information-intensive problems of the real world.

## ETHICS STATEMENT

The failures in planning and synthesis that we identified suggest a critical need for new methods that improve an agent's ability to form efficient, abstract reasoning plans and to maintain logical coherence in the face of plausible but irrelevant information.

## LIMITATIONS

MIRAGE is a first step, and we acknowledge its limitations. Our evaluation focuses on four domains, and the human-curated reasoning paths represent just one of many potential solution trajectories. Future work could expand the benchmark to more domains and languages, and explore more sophisticated automated metrics for evaluating the quality of reasoning paths.

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

# Appendices

## A  LLM USAGE

We employ Large Language Models (LLMs) to polish the writing and generate parts of datasets (§3.2.2). The design of the data collection pipeline, dataset statistics, experimental setup and execution, as well as the analysis of results, was carried out entirely by the authors without assistance from LLMs.

## B  EXPERIMENT DETAILS

### B.1  MODELS DETAILS

| Model | Architecture | Size | Version |
|---|---|---|---|
| BM25 (Robertson & Zaragoza, 2009) | Sparse | - | BM25 |
| BGE (Chen et al., 2024) | Encoder | 569M | BGE-m3 |
| E5-base (Wang et al., 2022) | Decoder | 109M | E5-base-v2 |
| SBERT (Reimers & Gurevych, 2019) | Encoder | 109M | all-mpnet-base-v2 |
| Inst-XL (Su et al., 2023) | Encoder | 1.5B | instructor-xl |
| Qwen-Embd (Zhang et al., 2025) | Encoder | 4B | Qwen-Embd-4B |
| GritLM (Muennighoff et al., 2024) | Encoder | 7B | GritLM-7B |
| ReasonIR Shao et al. (2025a) | Decoder | 8B | ReasonIR-8B |
| Qwen-2.5 (Team, 2024) | Decoder | 3B, 7B | Qwen-2.5-3B,Qwen-2.5-7B |
| Qwen-3 Team (2025) | Decoder | 14B | Qwen-2.5-14B |
| Llama-3 (AI@Meta, 2024) | Decoder | 3B | Llama-3-3B |

### B.2  MACHINE DETAILS

We used two machines for the scheme tested in this work: a machines with RTX 4090 and a machine with a single A100 80GB.

### B.3  STATISTICAL SIGNIFICANCE TEST

To prove that the performances are statistically significant, we conducted a T-test on two of our baselines: R-Search and Search-R1. Below is the result.

| Model | Legal | Finance | Technology | Academia |
|---|---|---|---|---|
| t-statistic | 19.8 | 16.6 | 19.6 | 17.4 |
| p-values | 0.00236 | 0.00172 | 0.00157 | 0.00274 |

## C   HUMAN VERIFICATION GUIDELINES

For questions in the benchmark, the reviewers are instructed to follow the following guidelines:

- Is the question general enough to include topics of all the sub-queries?

- Is there a logical or semantic relationship between the sub-queries and the question

For answers in the benchmark, the reviewer is instructed to follow the following guidelines:

- Is the answer detailed enough to include the details from all the reasoning.

- Does the answer a summarization of the reasoning.

For Adversarial Distractors, the reviewers are instructed to follow the following guidelines:

- Is the topic related to the sub-queries?

- Is this document a text-based document?

- Is the source creditable? (e.g. Wikipedia, Materials from univerisities, etc.)

## D   DATA EXAMPLES: ACADEMIA

**Question:** What is the nature of research roles at major technology companies like Microsoft, Apple, and Intel, and how do they compare to academic research positions?

**Answer:** Major technology companies such as Microsoft, Apple, and Intel actively engage in research, but their approaches vary. Microsoft and Intel have dedicated research divisions—Microsoft Research and Intel Labs, respectively—that hire PhDs from both industry and academia, including researchers with joint faculty appointments at universities. These researchers publish fundamental work in peer-reviewed conferences, workshops, and journals, and the companies file numerous patents. Apple, while investing heavily in R&D, rarely publishes in academic venues but still hires PhDs to conduct product-focused research, such as developing future technologies like the iPhone. Unlike academic research, corporate research tends to be directed toward profitable applications, though some labs, like Microsoft Research, maintain a balance closer to pure scientific inquiry. Historically, labs such as Xerox PARC and Bell Labs have been renowned for their groundbreaking fundamental research. The distinction between corporate research scientists and professors lies in their primary focus; while professors are evaluated heavily on their research contributions, corporate researchers align their work with business goals, though both roles emphasize research expertise.

---

**Multihop interleaving reasoning and retrieval**

**Turn 1**

**Sub-query 1:** Are people doing research for Microsoft, Apple, Intel experienced researchers/scientists or just experienced programmers using other people's research? Do they hire full time researchers for their research or do they take experienced/skilled professionals who are familiar working with java, C++ and other commercial technologies?

**Intermediate answer 1:** Microsoft has a prominent research arm, Microsoft Research. Intel has Intel Labs. Both organizations hire research-focused PhDs from both industry and academia, and both employ researchers that have joint appointments on research university faculty. Employees at both organizations also routinely publish fundamental research at peer-reviewed conferences and workshops, and in journals. Apple is a tougher nut to crack – they spend truckloads of money on R&D, but rarely publish in peer-reviewed venues. However, they do hire plenty of PhDs who do focused research on products that sometimes turn into the next iPhone. All three companies file many patents, as well. Many other big companies also have research labs, and openly hire researches specifically working on fundamental research. However, as with any company, research tends to be focused in the direction that could eventually produce profitable products – if you are considering a research job at a company, you should take this into consideration. For what it's worth, Microsoft Research seems to be closer to the research for the sake of science than some other labs, although there are others (notably, labs such as (formerly) Xerox PARC and Bell Labs are famous for groundbreaking fundamental research).

**Evidence 1:** Search *[Metropolis_(1927_film)]*

**Turn 2**

**Sub-query 2:** The reference you have did not mention professors at all $research scientist professor$.

**Intermediate answer 2:** sorry my mistake.But are'nt they almost the same thing? One of the most important things when hiring a professor is what research he has done

**Turn 3**

**Sub-query 3:** I should add that getting a job at Microsoft Research is comparable to and as rigorous as getting a job as a professor at a top university.

**Intermediate answer 3:** The linked question about job interviews isn't in any way about the difficulty of getting hired, but rather about how much they hassle you during the interview (brain teasers, details about programming languages, etc.).

## E  DATA EXAMPLES: TECHNOLOGY

**Question:** What are the key considerations and steps for provisioning and configuring SAP HANA on IBM Cloud Virtual Servers for VPC, including storage profiles, backup solutions, and deployment methods?

**Answer:** To provision and configure SAP HANA on IBM Cloud Virtual Servers for VPC, several key steps and considerations must be addressed. First, prepare configuration and script files from the IBM-Cloud GitHub repository,..., Throughout these processes, note that IBM provides no warranty or technical support for these materials, and issues should be reported via GitHub.

**Multihop interleaving reasoning and retrieval**

### Turn 1

**Sub-query 1:** What are the steps to provision a virtual server for SAP HANA with the required storage configuration on IBM Cloud Virtual Servers for VPC?

**Intermediate answer 1:** Sure, I can guide you through the process. Here are the steps: 1. Prepare your configuration and script files. You can find these in the GitHub repository [https://github.com/IBM-Cloud/sap-hana-single-vsi](https://github.com/IBM-Cloud/sap-hana-single-vsi)...If you encounter any issues or have any questions, you can open a GitHub issue.

### Turn 2

**Sub-query 2:** What storage profiles are recommended for SAP HANA on IBM Cloud Block Storage for Virtual Private Cloud?

**Intermediate answer 2:** For SAP HANA on IBM Cloud Block Storage for Virtual Private Cloud, it's recommended to use storage profiles that provide high IOPS per gigabyte (IOPS/GB)...For an overview of all available storage profiles, you can refer to the [VPC Block Storage Profiles](https://cloud.ibm.com/docs/vpc?topic=vpc-block-storage-profiles) documentation.

### Turn 3

**Sub-query 3:** How can I create a backup of my SAP HANA database on IBM Cloud Object Storage (COS)?

**Intermediate answer 3:** To create a backup of your SAP HANA database on IBM Cloud Object Storage (COS), you can follow these general steps: 1. First, you need to manually deploy a VPC and configure a backup/restore on iCOS with a backint hana agent. This can be time-consuming, but it ensures that your backup process is standardized and less prone to error. 2. Terraform and Ansible can be used for automating the deployment processes. The Terraform automation, in particular, assures a much quicker implementation. 3. Once your VPC is deployed and the backup/restore on iCOS is configured, you can use the backint hana agent to create a backup of your SAP HANA database.

### Turn 4

**Sub-query 4:** What are the steps to deploy SAP BW/4HANA release on dedicated SAP HANA 2.0 BOX VSI using Terraform and Ansible?

**Intermediate answer 4:** To deploy the SAP BW/4HANA release on dedicated SAP HANA 2.0 BOX VSI using Terraform and Ansible, you can follow these steps: 1. Prepare your configuration files. You can find these in the GitHub repository [https://github.com/IBM-Cloud/sap-bw4hana](https://github.com/IBM-Cloud/sap-bw4hana). You'll need to modify the 'input.auto.tfvars' file to customize the resources for your solution. 2. Enter the floating IP and subnet information from the Bastion server. 3. By default, the VSI is configured with Red Hat Enterprise Linux® 7.x for SAP Applications (amd64), two SSH keys that are configured to access as root user on SSH, and five storage volumes. 4. You can change the default settings to match your solution. You also specify the location where you downloaded the SAP kits. 5. The IBM Cloud Provider plug-in for Terraform on IBM Cloud uses these configuration files to provision a VPC in your IBM Cloud account. 6. After the VPC is created, you can use Ansible to automate the deployment process. Please note that there is no warranty or technical support available for these materials from IBM. If you encounter any issues or have any questions, you can open a GitHub issue.

### Turn 5

**Sub-query 5:** What are the KPIs defined by SAP HANA for the storage configurations of a single-node system?

**Intermediate answer 5:** To fulfill the KPIs defined by SAP HANA for the storage configurations of a single-node system, each profile needs different storage volumes. These configurations are mandatory storage configurations, not sample storage configurations, because they are the tested and certified storage layouts that comply with SAP HANA Tailored Data Center Integration (TDI) Phase 5. It is highly recommended to stick to these specific specifications. The specific storage configurations depend on the profile and are listed in detail in the SAP HANA TDI Phase 5 documentation. Unfortunately, this document is not available to me. Therefore, I'm unable to provide the specific storage configurations. However, you can refer to the SAP HANA TDI Phase 5 documentation for this information.

# F    DATA EXAMPLES: FINANCE

**Question:** What are the key considerations and rules regarding ISA allowances, transfers, and savings strategies?

**Answer:**  The key considerations and rules regarding ISA allowances, transfers, and savings strategies encompass several important aspects. Firstly, there are significant disadvantages to having someone else save for you using their ISA allowance, including the depletion of their tax-free savings allowance, reliance on their willingness to return the money, potential financial risks if creditors target the ISA funds, tax liabilities on interest when the money is returned, and the possibility of illegal tax evasion if undeclared. Secondly, existing shares in ISAs do not affect future ISA allowances, as only new cash contributions utilize the annual allowance, allowing you to open new ISAs for future savings. The annual ISA allowance is £20,000, which can be allocated across different ISA types, with a £4,000 limit for Lifetime ISAs. It is advisable to keep some savings in cash for liquidity, especially for short-term needs, to avoid selling investments at unfavorable times. Transfers between ISA types are permitted if the new ISA accepts transfers, but withdrawing and redepositing funds can lead to tax complications. If an ISA's interest rate becomes uncompetitive, transferring directly to a better-performing ISA is recommended. However, specific details about combining Help to Buy ISAs with Stocks and Shares ISAs in the same tax year are not available. Overall, effective ISA management involves understanding allowance limits, transfer rules, and strategic savings approaches to optimize tax benefits and financial flexibility.

**Multihop interleaving reasoning and retrieval**

**Turn 1**

**Sub-query 1:** What are the disadvantages of getting someone else to save for you with their ISA allowance?
**Intermediate answer 1:** The disadvantages of having someone else save for you with their ISA allowance include using up their tax-free savings allowance, relying on them to voluntarily give you the money back, potential financial problems for the person holding the ISA causing creditors to go after the money, tax liability on the interest when the money is returned to you, and the risk of illegal tax evasion if you don't declare the income. Overall, it's not an effective way to bypass any restrictions on your ISA holdings.

**Turn 2**

**Sub-query 2:** What happens to my existing shares in their existing ISAs in relation to my future ISA allowances?
**Intermediate answer 2:** Your existing shares in their existing ISAs do not impact your future ISA allowances. Only new cash paid into an ISA account utilizes your ISA allowance. You can leave your existing shares in their existing ISAs and open new ISAs for future contributions that suit your current plans.

**Turn 3**

**Sub-query 3:** How much can I save in ISAs in one tax year?
**Intermediate answer 3:** You can save up to £20,000 in one type of account or split the allowance across some or all of the other types. You can only pay £4,000 into your Lifetime ISA in a tax year. For example, you could save £11,000 in a cash ISA, £2,000 in a stocks and shares ISA, £3,000 in an innovative finance ISA and £4,000 in a Lifetime ISA in one tax year.

**Turn 4**

**Sub-query 1:** Should I keep some savings in cash?
**Intermediate answer 1:** It is advisable to consider keeping some of your savings in cash, for example in a current account that pays decent interest on the balance. This is especially important if you might need to dip into your savings for things like holidays and a car, as selling shares when the market is low can be risky.

**Turn 5**

**Sub-query 4:** Can I transfer money from one type of ISA to another?
**Intermediate answer 4:** Yes, you can transfer money from one type of ISA to another, as long as the new ISA allows transfers in. It's important to transfer the money directly between ISAs, and not to take the money out and then try to pay it into another ISA, as this can have negative tax implications.

**Turn 6**

**Sub-query 5:** What should I do with my old ISA if the interest rate becomes less competitive over time?
**Intermediate answer 5:** If the interest rate on your old ISA becomes less competitive over time, you should check if you can get a better rate in a different ISA that allows transfers in. If so, you should arrange to make the transfer directly between ISAs, as taking the money out and then trying to pay it into another ISA can have negative tax implications.

## G   DATA EXAMPLES: LEGAL

**Question:** What are the legal implications of downloading content from YouTube, and how do YouTube's Terms of Service and copyright laws apply?
**Answer:** Downloading content from YouTube is not inherently a criminal act, but it typically breaches YouTube's Terms of Service, which users agree to when accessing the platform. According to YouTube's Terms of Use (specifically section 5B), downloading content is prohibited unless YouTube explicitly provides a download link for that content. This breach of contract could expose users to legal action from YouTube, including claims for lost advertising revenue or punitive damages. Legally, the permissibility of downloading depends on jurisdiction, with most regions allowing personal copies

of purchased media but restricting online content under service terms. Content owners retain rights, licensing reproduction and distribution to YouTube, meaning users lack rights to distribute or modify the content unless explicitly permitted. Fair Use or Fair Dealing defenses may apply, depending on factors like the purpose of use, the nature of the copyrighted work, the amount used, and the effect on the work's market value. However, without specific case law, the legality largely hinges on YouTube's Terms and copyright principles. Additionally, using third-party tools like youtube-dl could still bind users to YouTube's Terms if the tool accesses YouTube's API. While end users are rarely sued for downloading, creating or distributing software for downloading YouTube content could also lead to legal challenges from Google.

### Multihop interleaving reasoning and retrieval

#### Turn 1

**Sub-query 1:** I've always been under the assumption that doing such was illegal. I don't see any difference between downloading music from YouTube and downloading music from The Pirate Bay. Apparently, there is this Software Recs SE question that talks precisely about downloading from YouTube. Related: Piracy/File sharing - Why aren't songs, movies or ebooks given for free (+ads) like TV?

**Intermediate answer 1:** It's not a crime per se, but you're breaching contract if you're accessing it by normal means, that is, through a Web browser or through the API. Youtube Terms of Use 5B, emphasis added: Content is provided to you AS IS. You may access Content for your information and personal use solely as intended through the provided functionality of the Service and as permitted under these Terms of Service. ... Nature and purpose of the use Nature of the copyrighted work Amount of the copyrighted work used The effect of the use on the value of the copyrighted work In the absence of case law to the contrary[1], the Terms of Use, copyright law, and fair use law are only legal principles we can rely upon in determining the legality of downloading content from Youtube. 1. It's possible that there is case law, but I've just spent a bit of time searching and haven't found any cases where end users were sued for downloading content.

#### Turn 2

**Sub-query 2:** Would I ever be legally considered to have agreed to these terms of service, if all I did was entering 'youtube-dl https://www.youtube.com/watch?v=XXX' into my command line?

**Intermediate answer 2:** Clause 1A: By using or visiting the YouTube website or any YouTube products, software, data feeds, and services provided to you on, from, or through the YouTube website (collectively the S̈ervice) you signify your agreement to (1) these terms and conditions **or** If a command-line tool uses the Youtube API to access the Content, then the user who generated the API key is bound by the Youtube API Terms.

#### Turn 3

**Sub-query 3:** It is not banned by criminal law, but it is a breach of YouTube's contract. You cannot be sentenced to prison, but you can be sued by Google.

**Intermediate answer 3:** I understand that you can be sued by Google for downloading videos, but do you think you could be sued by Google for just making available a software or an online application for downloading videos?

## H    SIDE BY SIDE COMPARSION

Question: **What are the financial implications and strategic considerations of self-insuring versus purchasing insurance for risk management?**

| Type | Model Thinking Path | Human Thinking Path |
|---|---|---|
| Thinking | I need to understand the financial implications and strategic considerations of self-insuring versus purchasing insurance for risk management. I'll search for it. | Why is self-insuring a fleet considered capital intensive? |
| Query | I need to understand the financial implications and strategic considerations of self-insuring versus purchasing insurance for risk management. I found out that self-insurance involves setting aside funds to cover potential losses, while purchasing insurance involves paying premiums to a third-party provider. I'll search for the financial implications and strategic considerations now. | How do insurance companies negotiate better rates? |
| Query | I need to find the financial implications and strategic considerations of self-insuring versus purchasing insurance for risk management. I found out that self-insurance involves setting aside funds to cover potential losses, while purchasing insurance involves paying premiums to a third-party provider. I found out that self-insurance can result in cost savings, tailored coverage, and investment potential, but it also carries financial risk, legal requirements, and the need for financial discipline. Purchasing insurance provides protection from catastrophic loss, better rates on dealing with recovering from losses, and potentially saves more than the administrative overhead and profit taken off the top. Now I can provide the answer. | What is the impact of self-insuring on liquidity? |
| Query | | How does the value of insurance predictability decrease as the value of coverage decreases? |

| | | |
|---|---|---|
| Query | | Why is it easier to self-insure for minor losses? |
| Query | | If I were to invest in a diversified portfolio, what would be the benefit? |
| Query | | How does spreading risk out benefit individuals and businesses? |
| Answer | cost savings | Self-insuring is considered capital intensive because it requires setting aside a significant amount of money to cover potential risks, which can tie up funds and impact liquidity. This approach banks on the assumption that the total cost of risks will not exceed the reserved funds, but if risks are significant, it demands a large upfront investment. In contrast, purchasing insurance spreads risk across a larger group, offering more manageable payments and predictable costs, which aids financial planning and preserves liquidity. Insurance companies leverage their large customer base, extensive risk data, and established relationships with service providers to negotiate better rates, making insurance a cost-effective option. The value of insurance predictability diminishes for minor losses, as their financial impact is less severe, making self-insurance more feasible for such cases. The Law of Large Numbers underpins insurance by ensuring that as the number of policyholders grows, the actual claims ratio aligns with the expected one, allowing insurers to charge affordable premiums. Ultimately, spreading risk through insurance provides financial stability and protection against catastrophic losses, benefiting both individuals and businesses by pooling resources and sharing the financial burden. |