# OpenReview forum: "MIRAGE: MULTI-HOP INTERLEAVED REASONING AND RETRIEVAL-GROUNDED EVIDENCE"
_ICLR.cc/2026/Conference — Submitted to ICLR 2026_

### Official Review · Reviewer_WyBe · 2025-10-29

**Soundness:** 2
**Presentation:** 2
**Contribution:** 2
**Rating:** 2
**Confidence:** 4

**Summary:**

This paper introduces MIRAGE, a benchmark designed to evaluate llm-based RAG systems on the interleaved task of multi-hop retrieval and evidence-grounded reasoning.
The authors claim that this interleaved RAG capabilities is critical for real-world problem-solving, but this spectrum is not well addressed by existing benchmarks.
Therefore, this paper propose MIRAGE, which consists of 579 instances across four domain (Legal, FInance, Tech, Academy). Both of them needs llms to generate sub-queries iteratively, retrieve knowledge from adversarial corpus, to find the final answer.

The authors conduct experiments comparing closed-book LLMs, single-turn RAG, multi-hop RAG, and interleaved RAG systems, finding that even SOTA RAG frameworks achieve at most 61.5% Fact-Score, with failures attributed to premature reasoning and distractor vulnerability.
The paper positions MIRAGE as a public resource to advance agentic reasoning capabilities in future research.

**Strengths:**

1. This paper introduces the MIRAGE benchmark for Retrieval-Augmented Generation (RAG), designed to address challenges such as interleaved retrieval and noise resistance in the era of AI agents. The benchmark focuses on comprehensive evaluation of the retrieval-augmented workflow, assessing not only the accuracy of final answers but also capabilities like robustness to retrieval interference. It also provides relevant environments with both standard and adversarial settings, ensuring a high degree of reproducibility.

2. The authors evaluated current academic and industrial models and products on MIRAGE, drawing preliminary conclusions from the results.

**Weaknesses:**

1. What are the key distinctions between the benchmark proposed in this paper and previous or subsequent RAG benchmarks?
    - Pre-agent era benchmarks: Multi-hop QA datasets such as HotpotQA and MuSiQue.
    - Agent-era benchmarks: Web browsing benchmarks represented by BrowseComp.
    - Both types of benchmarks involve multi-turn, interleaved retrieval and generation, making them suitable for evaluating agent-era reasoning models. Moreover, some multi-hop QA benchmarks evaluate not only the correctness of the final answer but also the accuracy of intermediate retrieval steps.

2. The proposed benchmark has a relatively small sample size, containing only 500+ instances, which limits its applicability to evaluations in narrow domains.

3. The benchmark lacks sufficient difficulty. A high-quality benchmark should possess foresight by focusing on challenges that current models struggle to solve. For example, when BrowseComp was introduced, most existing methods achieved only single-digit scores. In contrast, the benchmark discussed here appears to lack challenge, as some models already achieved scores above 60 at its release. This raises the question of whether the benchmark fails to provide meaningful guidance for future model optimization due to its limited difficulty.

**Questions:**

please see weakness section for my quesiton.

---

> ### Author Response · Authors · 2025-11-25
> **Response to Weakness 1**
>
> We sincerely thank the reviewer for the thoughtful and constructive comments. MIRAGE introduces a fundamentally new task formulation that differs significantly from existing RAG benchmarks. The key distinctions are as follows:
>
> 1. **Sub-reasoning–intensive retrieval generation:**
> Each query in MIRAGE is designed to be decomposable into multiple interdependent sub-queries. These sub-queries must be generated based on the outcomes of earlier retrieval or reasoning steps. This structure requires models to perform iterative reasoning, rather than simply issuing a single retrieval call or following a fixed-hop pipeline.
>
> 2. **End-to-end answer construction leveraging the entire reasoning trajectory:**
> In contrast to both pre-agentic and agentic-era benchmarks—where the final answer often depends primarily on the last retrieval step—MIRAGE requires models to integrate evidence and intermediate conclusions accumulated across all prior sub-reasoning steps. This design enforces deeper reasoning continuity and multi-step synthesis.
>
> To make these distinctions clearer, we include a trajectory comparison illustrating how MIRAGE differs fundamentally from the benchmarks referenced by the reviewer.
>
> ```
> # HotpotQA
>
> Q: Which magazine was started first Arthur's Magazine or First for Women?
>
> Trajectory:
> Search 1: Start Date of Arthur's Magazine. -> Fact 1: 1844–1846 ->
> Seaerch 2: Start Date of First for Women -> Fact 2: The magazine was started in 1989 ->
> Reasoning 1: 1846 < 1989 -> Answer:  Arthur's Magzine
>
> # As you can see, the final answer only include information from the Reasoning 1
> # rather than incorporate Fact 1 and Fact 2.
>
> # BrowseComp
>
> Q: An African author tragically passed away in a tragic road accident.
>    As a child, he'd wanted to be a police officer.
>    He lectured at a private university from 2018 until his death.
>    In 2018, this author spoke about writing stories that have no sell by date in an interview.
>    One of his books was selected to be a compulsory school reading in an African country in 2017.
>    Which years did this author work as a probation officer?
>
> Trajectory:
> Search 1: Which book was selected to be a compulsory school reading in an African country in 2017. -> Fact 1: Author names. -> Search 2: Which author spoke about writing stories that have no sell by date in an interview in 2018. -> Fact 2: Author names. -> Search 3: Author lectured at a private from 2018 to his death-> Fact 3: ... -> Reasoning 1: Compare Fact 1, Fact 2, Fact 3 ... Get the final answer. -> Answer: xxx
>
> # The final answer is generated by Reasoning 1 instead of incorporating all the facts into the final answer.
>
>
> # MIRAGE
>
> Q: What are the key considerations and steps for provisioning and configuring SAP HANA on IBM Cloud Virtual Servers for VPC, including storage profiles, backup solutions, and deployment methods?
>
> Trajectory:
> Search 1: What are key considerations for configuraing SAP HANA? ->
> Reasoning 1: The considertions are xxxx. ->
> Search 2: What about setting up the storage drives? ->
> Reasoning 2: The steps are xxx.
> -> .... ->
> Final answer: There are serval key considerations and steps for setting up SAP HAMA on IBM Cloud VR for VPC. (Information from Reasoning1) (Information from Reasoning 2) (Information from Reasoning 3) ...
>
> # The final answer is a summarization of all the reasoning from previous steps rather than from the last reasoning step.
> ```
> 3. **Reasoning Steps Aimed at Evidence Expansion, Not Just Query Refinement: **
> The purpose of intermediate reasoning in MIRAGE is to uncover and synthesize additional supporting information, rather than merely refine a query toward a single target document. This encourages broader evidence gathering and deeper reasoning, which is critical for realistic multi-hop information-seeking tasks.

---

> ### Author Response · Authors · 2025-11-25
> **Response to Weakness 2**
>
> We thank the reviewer for this thoughtful comment.
>
> 1. **Benchmark Size Comparison:**
> MIRAGE contains 579 instances with an average of 4 reasoning hops per instance, resulting in over 2,000 single-hop queries in total. In comparison, BrowseComp includes a little over 1,200 single-hop questions. Thus, in terms of the effective number of queries, our benchmark is comparable tosimilar existing benchmarks, even though the number of top-level instances may appear smaller at first glance.
>
> 2. **Representativeness and Data Quality**
> In addition, the 579 samples in MIRAGE is **highly representative** of real-world relevance designed to capture the authentic distribution of user queries and the retrieval challenges that arise in practical scenarios. The` high quality of MIRAGE is guranteed by the complexity of sythetic end-to-end QA pairs, diversity of retrieval corpus and rigorous construction of manual verification and refinement` spanning diverse subdomains and difficulty levels, avoiding bias toward trivial cases. Thus, despite the smaller size, the experiement results in our paper provides **actionable insights into model capabilities for real-world multi-hop problems**.
>
> 3. **Scalable Data Collection Pipeline:**
> More importantly, our work not only introduces the benchmark itself but also proposes a **scalable data collection pipeline** (see Section 3.2) that can be readily extended to other domains. This pipeline enables `continuous data expansion and adaptation ensuring long-term scalability`, allowing us to systematically incorporate new domains and query types over time.
>
> We acknowledge that larger datasets are valuable for broad evaluations, but our focus here is on depth (multi-hop reasoning and integrating multiple reasoning steps) and real-world fidelity—priorities that justify the current sample size. Future work will include incremental expansions via the **LiveBench** pipeline.

---

> ### Author Response · Authors · 2025-11-25
> **Response to Weakness 3**
>
> We appreciate the reviewer’s insightful feedback regarding the difficulty of our benchmark. Below, we clarify our design rationale and address the concerns:
> 1) **Simiar to real-world questions**
> Unlike BrowseComp (which focuses on artificially designed, highly complex tasks), our benchmark aims to evaluate `whether current models and agents can solve real-world multi-hop problems`. Our tasks are curated to reflect practical scenarios (e.g., StackOverflow troubleshooting), where difficulty arises naturally from the need to integrate information across multiple sources and reasoning steps, rather than from artificial constraints.
> 2) **Certain models are trained to be effective**
> Newer models and agentic pipelines—such as GPT-4 and GPT-5–based systems like DeepSearch—have been fine-tuned on reasoning-intensive, retrieval-based benchmarks such as BrowseComp. As a result, these models are naturally expected to perform well on benchmarks with similar characteristics, including MIRAGE. To better isolate the benchmark’s difficulty and avoid conflating results with recent model-specific training, we conducted an additional evaluation using older models. As shown in the table below, even when controlling for retrieval quality, older models perform noticeably worse than newer models trained on similar tasks. This further demonstrates that high performance on MIRAGE is not trivial and depends on advanced reasoning and retrieval capabilities.
>
> | Model              | FS | STR-EM | BS | nDCG@10 | Pre. | Rec. |
> |--------------------|----|--------|----|---------|------|------|
> | o3+Thinking        |51.1| 49.1   |23.5| 22.1    | 21.1 |  21.5|
> | GPT-3.5-turbo      |47.8| 45.0   |20.7| 22.1    | 21.1 |  21.5|

---

> ### Author Response · Authors · 2025-11-28
> **Looking Forward for Your Responses**
>
> Dear Reviewer WyBe,
>
> We truly appreciate the time and effort you have dedicated to reviewing our work, and we sincerely appreciate your recognition of our strengths. We have done our best to address your comments by providing additional clarifications and examples, and we want to ensure that our responses and analyses fully address your concerns.
>
> If you have any further questions or require additional information from us, we would be more than happy to provide it. Your feedback is extremely valuable to us, and we want to make sure we have fully addressed all points raised.
>
> Thank you again for your thoughtful review and for taking the time to consider our responses. We sincerely appreciate your help in improving our work.
>
> Warm regards,
> Authors

---

### Official Review · Reviewer_VW56 · 2025-10-31

**Soundness:** 3
**Presentation:** 3
**Contribution:** 2
**Rating:** 6
**Confidence:** 4

**Summary:**

This paper introduces MIRAGE, a benchmark designed to evaluate multi-hop interleaved reasoning and retrieval-grounded evidence generation. It aims to simulate realistic RAG workflows where reasoning and retrieval alternate across multiple steps. The dataset covers several specialized domains (finance, legal, technology, academia) and includes human verification and adversarial distractor construction. Experiments benchmark various LLM-based RAG systems under single-hop, multi-hop, and interleaved settings, accompanied by detailed error analyses distinguishing planning vs. synthesis failures.

**Strengths:**

1. The paper targets a timely and practically relevant problem: evaluating RAG systems that require iterative retrieval and reasoning.
2. The benchmark is well-structured, with clear design motivation, adversarial distractor generation, and human verification steps.
3. Experiments are comprehensive, covering multiple reasoning configurations and offering useful error analyses (e.g., planning vs. synthesis failures).
4. The paper is clearly written and easy to follow, with reproducible methodology and well-presented results.

**Weaknesses:**

1. The paper’s identified gap: evaluating interleaved reasoning and retrieval, feels primarily conceptual rather than substantive. Similar processes have been studied in prior RAG and agentic frameworks (e.g., ReAct, RARR). MIRAGE’s contribution lies more in its framing and benchmark packaging than in a fundamentally new task formulation.
2. Given that MIRAGE emphasizes multi-hop reasoning, the evaluation could better reflect step-level performance (e.g., scoring per hop, planning accuracy, intermediate retrieval relevance). The current setup uses rather standard metrics and separates "reasoner" and "retriever" evaluation without integrating them into a unified process-level metric. This limits the benchmark’s ability to assess true multi-hop reasoning quality. Also, there's a typo in Table 6, Reasoner instead of Resoner.

**Questions:**

1. The core task of interleaved reasoning and retrieval appears conceptually similar to prior agentic frameworks (e.g., ReAct). Could the authors please elaborate on what substantively differentiates the MIRAGE task formulation from these existing paradigms? Is the novelty primarily in the challenging, domain-specific corpus and the adversarial distractors, or is there a fundamental difference in the required reasoning process itself?
2. Given that MIRAGE is explicitly designed to evaluate the interleaved process of multi-hop reasoning, the evaluation in Table 6  seems focused on end-to-end performance (Fact-Score, nDCG@10) rather than the quality of the intermediate steps. Could the authors comment on why process-level metrics (e.g., planning accuracy of sub-queries, or hop-by-hop retrieval relevance) were not included? How can the benchmark effectively diagnose "Planning Failures" without such metrics?

---

> ### Author Response · Authors · 2025-11-25
> **Response to Question 1**
>
> We thank the reviewer for raising this important point regarding the novelty of our work. Our contributions center on both the complexity of the task and the necessity of a comprehensive reasoning process. The novelty of our benchmark can be summarized in the following three aspects:
>
> 1. **Reasoning-Intensive Multi-Hop Retrieval Benchmark:**
> To the best of our knowledge, this is the first benchmark designed explicitly to evaluate reasoning-intensive, multi-hop retrieval in realistic, domain-specific settings. Unlike prior work that isolates retrieval or reasoning in simpler forms, our benchmark integrates both dimensions holistically.
> 2. **Evaluation Under Domain-Specific, Adversarial Conditions:**
> The benchmark is constructed to test model robustness under domain-specific, retrieval-intensive scenarios involving adversarial distractors. These distractors are drawn from the same corpus, requiring models to perform deliberate multi-step reasoning rather than superficial keyword matching.
> 3. **Provision of Detailed Reasoning Trajectories:**
> We include complete, curated reasoning trajectories that illustrate the intermediate reasoning and retrieval steps needed to solve each query. This facilitates analysis of step-level behavior and enables a more granular evaluation of planning, retrieval, and reasoning quality—an aspect largely absent in existing datasets.

---

> ### Author Response · Authors · 2025-11-25
> **Response to Question 2**
>
> We thank the reviewer for this thoughtful question. We initially considered incorporating step-level metrics (e.g., process-level retrieval accuracy) to measure intermediate reasoning quality. However, during preliminary analysis, we observed that `different models often follow distinct reasoning trajectories when solving the same query`. Consequently, aligning and evaluating each intermediate step across models becomes non-trivial and potentially misleading, as `models may reach the correct answer through different reasoning paths or retrieval strategies`.
>
> Furthermore, `step-level evaluation tends to be highly sensitive to the model’s training paradigm and reasoning style`. As we noted in the paper, human reasoning trajectories differ substantially from those of current LLMs, making direct human-aligned step-level comparisons less reliable for assessing model capability.
>
> To address this concern, we have conducted a more **detailed analysis of step-wise retrieval relevance and planning accuracy across the intermediate reasoning steps** of different baseline models to illustrate their intermediate behaviors as below.
>
> **Standard RAG**
> | Model              | Avg. Planning Rec. | Avg. Step Ret Rec*   |
> |--------------------|--------------------|----------------------|
> | Qwen-3             | 25.0               | 11.8                 |
> | Qwen-3+Thinking    | 26.1               | 11.8                 |
> | OpenAI o3          | 26.3               | 11.8                 |
> | OpenAI o3+Thinking | 26.4               | 11.8                 |
> | DeepSeek V3        | 26.1               | 11.8                 |
> | DeepSeek R1        | 26.4               | 11.8                 |
>
> **Multi-Hop RAG**
> | Model              | Avg. Planning Acc. | Avg. Step Ret Recall |
> |--------------------|--------------------|----------------------|
> | Self-RAG           | 28.6               | 29.8                 |
> | SeakR              | 28.4               | 26.4                 |
> | Multi-hop RAG      | 24.8               | 13.8                 |
>
> **Interleaving RAG**
> | Model              | Avg. Planning Acc. | Avg. Step Ret Recall |
> |--------------------|--------------------|----------------------|
> | Search-R1          | 33.6               |41.6                  |
> | R-Search           | 33.4               |39.9                  |
> | Re-Call            | 32.8               |36.7                  |
>
> *Planning recall is calculated by measuring if a reasoning step shows up in the groundtruth reasoning steps: $r_i \in r_{gt}$*
> *Step Retrieval Recall is calculated by measuring the recall at each reasoning step against the total retrieved document.*
>
> We observed that both Simple-RAG and Multi-Hop RAG exhibit lower step-level Retrieval Recall, largely due to poor retrieval performance at specific steps. In several cases, individual steps achieve a recall of 0 because the model’s generated sub-query is semantically misaligned with the corresponding ground-truth sub-query. This mismatch prevents the model from retrieving any relevant evidence at those steps, even when it performs reasonably well on the overall question.
>
> In contrast, the Interleaving-RAG paradigm is less affected by this issue because it employs a dynamic query generation mechanism that adapts the retrieval query based on previously retrieved evidence. This allows it to produce more effective sub-queries throughout the reasoning process and leads to higher step-level recall.
>
> Regarding planning recall, the scores are also relatively low across models. This is primarily due to the divergence between human-designed reasoning trajectories and model-selected trajectories. Different models often choose alternative reasoning paths that deviate from the annotated ground-truth plan, resulting in lower alignment scores even when the overall answer may still be correct.

---

### Official Review · Reviewer_mKar · 2025-11-01

**Soundness:** 3
**Presentation:** 4
**Contribution:** 3
**Rating:** 6
**Confidence:** 3

**Summary:**

This paper introduces MIRAGE, a benchmark designed to evaluate LLMs' ability to dynamically interleave multi-hop retrieval and evidence-grounded reasoning. The benchmark comprises 579 tasks across four domains (Legal, Finance, Technology, Academia), constructed from real-world conversational data through a three-stage pipeline: (1) sourcing conversational seeds, (2) synthesizing complex reasoning tasks, and (3) constructing adversarial retrieval corpora. Comprehensive experiments reveal a stark performance gap that the best academic system achieving only 61.5% task success, with interleaved RAG agents significantly outperforming single-turn and structured multi-hop approaches. The authors identify two primary failure modes: planning failures and synthesis failures.

**Strengths:**

1. The paper addresses a genuine evaluation gap. MIRAGE is designed to evaluate the interleaved process where agents must dynamically decide when to retrieve and what to search for.
2. The paper includes comprehensive evaluation across multiple architectures (single-turn, multi-hop, interleaved, commercial systems).
3. The paper is well-written with clear motivation, good use of examples, and visualizations.

**Weaknesses:**

1. 579 tasks is relatively small for a benchmark, especially when split across 4 domains.
2. The paper does not report statistical significance of the results.
3. Domain distribution is highly skewed (Legal: 216, Technology: 184, Academia: 110, Finance: 69). This imbalance may affect the generalizability of conclusions.

**Questions:**

1. BERTScore might be unsuitable for such kind of abstractive, evidence-grounded answers. For future papers, if the authors can only pick one metric to report, which one would the authors recommend?
2. Would it be feasible to conduct a human evaluation on some samples (e.g., 100 examples) to validate the automated metrics and provide qualitative insights?
3. Can you provide a breakdown of results by domain? Are certain domains systematically harder?
4. Are there any considerations for avoiding potential benchmark fitting in the future?
4. "disconnect" -> "disconnection" in line 52; "Soucing" -> "Sourcing" in Figure 1. "Resoner" -> "Reasoner" in Table 6.

---

> ### Author Response · Authors · 2025-11-25
> **Response to Weakness 1**
>
> We thank the reviewer for this thoughtful comment.
>
> 1. **Benchmark Size Comparison:**
> MIRAGE contains 579 instances with an average of 4 reasoning hops per instance, resulting in over 2,000 single-hop queries in total. In comparison, a similar benchmark, BrowseComp, includes a little over 1,200 single-hop questions. Thus, in terms of the effective number of queries, our benchmark is comparable to similar existing benchmarks, even though the number of top-level instances may appear smaller at first glance.
>
> 2. **Scalable Data Collection Pipeline:**
> More importantly, our work not only introduces the benchmark itself but also proposes a **scalable data collection pipeline** (see Section 3.2) that can be readily extended to other domains. This pipeline enables `continuous data expansion and adaptation ensuring long-term scalability`, allowing us to systematically incorporate new domains and query types over time.
>
> We acknowledge that larger datasets are valuable for broad evaluations, but our focus here is on depth (multi-hop reasoning and integrating multiple reasoning steps) and real-world fidelity—priorities that justify the current sample size. Future work will include incremental expansions via the **LiveBench** pipeline.

---

> ### Author Response · Authors · 2025-11-25
> **Response to Weakness 2**
>
> We thank the reviewer for this valuable suggestion. We agree that reporting statistical significance is crucial for assessing the robustness of the results.
>
> We conducted a paired t-test on the FactScore of two of the baselines, R-Search and Search-R1, as they both use Qwen-2.5-7B as backbone model.
>
> | Model       | Legal   | Finance | Technology | Academic |
> |-------------|---------|---------|------------|----------|
> | p-value     | 2.36*10^-36 |1.72*10^-35  |1.57*10^-36 |2.74*10^-36   |
>
> As shown in the table, all four domains yield p-values below 0.05, indicating that the observed performance differences are statistically significant and unlikely to be due to random variation.
>
> We have added these results to the revised manuscript and will explicitly state the statistical testing procedure and significance levels in the appendix to enhance the rigor and interpretability of our findings.

---

> ### Author Response · Authors · 2025-11-25
> **Response to Weakness 3**
>
> We thank the reviewer for this thoughtful comment. To ensure that the imbalance does not affect the generalizability of the conclusion, we compared the results of various models by the domain.
>
> **Legal**
> | Model              | FS | STR-EM | BS | nDCG@10 | Pre. | Rec. |
> |--------------------|----|--------|----|---------|------|------|
> | Qwen-3             |38.9|34.5    |19.7| 21.4    |20.3  |20.4  |
> | Qwen-3+Thinking    |41.8|35.6    |20.3| 21.4    |20.3  |20.4  |
> | OpenAI o3          |49.3|42.1    |22.5| 21.4    |20.3  |20.4  |
> | OpenAI o3+Thinking |49.5|48.9    |23.5| 21.4    |20.3  |20.4  |
> | DeepSeek V3        |49.6|41.0    |18.2| 21.4    |20.3  |20.4  |
> | DeepSeek R1        |49.7|39.9    |19.5| 21.4    |20.3  |20.4  |
> | Self-RAG           |58.6|41.3    |28.0| 40.1    |38.8  |39.6  |
> | SeakR              |57.6|41.0    |27.9| 36.5    |36.6  |36.5  |
> | Multi-hop RAG      |40.8|39.0    |20.3| 16.3    |16.5  |14.1  |
> | Search-R1          |59.1|40.3    |38.4| 42.4    |41.4  |43.2  |
> | R-Search           |58.6|39.6    |35.5| 40.7    |40.9  |41.5  |
> | Re-Call            |58.2|37.5    |33.7| 40.5    |40.6  |40.3  |
>
> **Technology**
> | Model              | FS | STR-EM | BS | nDCG@10 | Pre. | Rec. |
> |--------------------|----|--------|----|---------|------|------|
> | Qwen-3             |41.7|34.2    |19.4| 21.8    |20.5  |20.7  |
> | Qwen-3+Thinking    |42.8|35.5    |20.2| 21.8    |20.5  |20.7  |
> | OpenAI o3          |51.7|42.3    |22.7| 21.8    |20.5  |20.7  |
> | OpenAI o3+Thinking |51.8|49.0    |23.4| 21.8    |20.5  |20.7  |
> | DeepSeek V3        |52.0|41.1    |18.5| 21.8    |20.5  |20.7  |
> | DeepSeek R1        |51.2|41.0    |19.7| 21.8    |20.5  |20.7  |
> | Self-RAG           |60.5|41.2    |28.1| 39.7    |38.7  |39.7  |
> | SeakR              |59.9|39.9    |27.9| 36.7    |36.7  |37.0  |
> | Multi-hop RAG      |41.6|38.8    |20.4| 16.7    |16.3  |14.3  |
> | Search-R1          |61.6|41.1    |38.5| 42.1    |41.5  |43.7  |
> | R-Search           |60.6|41.0    |35.7| 40.9    |40.8  |42.1  |
> | Re-Call            |60.5|37.6    |33.8| 40.6    |40.7  |40.8  |
>
> **Academia**
> | Model              | FS | STR-EM | BS | nDCG@10 | Pre. | Rec. |
> |--------------------|----|--------|----|---------|------|------|
> | Qwen-3             |41.8|34.3    |20.2| 22.5    |21.5  |21.4  |
> | Qwen-3+Thinking    |42.1|35.8    |20.6| 22.5    |21.5  |21.4  |
> | OpenAI o3          |50.5|42.4    |22.8| 22.5    |21.5  |21.4  |
> | OpenAI o3+Thinking |50.6|49.2    |23.5| 22.5    |21.5  |21.4  |
> | DeepSeek V3        |50.8|41.2    |18.4| 22.5    |21.5  |21.4  |
> | DeepSeek R1        |51.3|41.1    |19.7| 22.5    |21.5  |21.4  |
> | Self-RAG           |61.5|41.3    |28.2| 39.9    |38.4  |39.2  |
> | SeakR              |61.6|41.1    |27.9| 40.0    |36.4  |36.5  |
> | Multi-hop RAG      |41.5|39.2    |20.4| 16.4    |16.1  |14.0  |
> | Search-R1          |62.2|40.6    |38.3| 42.2    |41.2  |43.6  |
> | R-Search           |62.0|39.7    |35.6| 40.6    |40.5  |41.7  |
> | Re-Call            |60.8|37.8    |33.6| 40.3    |40.3  |40.5  |
>
> **Finance**
> | Model              | FS | STR-EM | BS | nDCG@10 | Pre. | Rec. |
> |--------------------|----|--------|----|---------|------|------|
> | Qwen-3             |42.4|34.4    |20.3| 22.7    |22.1  |22.3  |
> | Qwen-3+Thinking    |42.5|36.3    |20.5| 22.7    |22.1  |22.3  |
> | OpenAI o3          |52.5|42.4    |22.8| 22.7    |22.1  |22.3  |
> | OpenAI o3+Thinking |52.5|49.3    |23.6| 22.7    |22.1  |22.3  |
> | DeepSeek V3        |52.4|41.5    |18.1| 22.7    |22.1  |22.3  |
> | DeepSeek R1        |50.6|40.8    |19.5| 22.7    |22.1  |22.3  |
> | Self-RAG           |60.6|41.4    |28.1| 39.5    |38.7  |39.1  |
> | SeakR              |57.3|42.0    |27.9| 36.6    |36.3  |37.2  |
> | Multi-hop RAG      |42.1|39.4    |20.1| 16.5    |16.3  |14.4  |
> | Search-R1          |63.1|40.0    |38.0| 42.5    |41.1  |43.9  |
> | R-Search           |60.4|38.9    |35.6| 40.8    |40.6  |42.3  |
> | Re-Call            |61.3|37.5    |33.7| 40.6    |40.4  |40.8  |
>
> As shown in the table, the trends match with our conclusion in the paper that Standard RAG scheme cannot directly solve the problem, and Multi-hop RAG scheme has better performance than Standard RAG scheme. Interleaving RAG provides the best pipeline performance.

---

> ### Author Response · Authors · 2025-11-25
> **Response to Question 1**
>
> We appreciate the reviewer’s suggestion regarding the use of a single primary metric for evaluation. This is indeed an important but challenging question, as there is currently no unified or widely accepted metric for evaluating evidence-based and open-ended question answering. Different metrics capture different aspects of model performance—such as factuality, semantic similarity, retrieval quality, or reasoning completeness—making it difficult for any single metric to fully represent the complexity of the task.
>
> In our work, we therefore recommend using the full set of metrics presented in the paper to obtain a comprehensive evaluation across these dimensions. However, if a single primary metric must be chosen, we would recommend FactScore, as it most directly assesses the factual correctness and evidence grounding of the generated answers, which aligns closely with the goals of MIRAGE.

---

> ### Author Response · Authors · 2025-11-25
> **Response to Question 2**
>
> We thank the reviewer for this valuable suggestion. Following the reviewer’s recommendation, we conducted a human evaluation on 100 randomly selected samples covering all four domains in our benchmark. The results are shown below:
>
> | Type      | Accuracy | Recall   |
> | --------  | -------- | -------- |
> | Human     | 61.3     | 43.6     |
> | Search-R1 | 61.5     | 43.6     |
>
> The results show strong alignment between human judgments and automated scores, confirming the effectiveness and reliability of the proposed evaluation metrics.

---

> ### Author Response · Authors · 2025-11-25
> **Response to Question 3**
>
> We thank the reviewer for this helpful suggestion. We provided a table of results by the specific domain below.
>
> | Domain     | FactScore | Reacll   |
> | --------   | --------- | -------- |
> | Legal      | 59.1      | 43.2     |
> | Technology | 61.6      | 43.7     |
> | Academia   | 62.2      | 43.6     |
> | Finance    | 63.1      | 43.9     |
>
> *Results are from Search-R1 and we noticed the trends applicable to all other results as well*
>
> In general, we observe that questions from the legal domain tend to be more challenging, as they often require models to perform multi-step reasoning and retrieval from prior case records and legal articles. In contrast, other domains such as technology involve more direct factual or contextual retrieval, which are relatively less complex. We believe this breakdown and accompanying examples provide a clearer understanding of the dataset’s structure and the varying reasoning demands across domains.

---

> ### Author Response · Authors · 2025-11-25
> **Response to Question 4**
>
> We thank the reviewer for raising this important point regarding potential benchmark fitting. We fully acknowledge that, as with any public benchmark, there exists a possibility that models may eventually become overfitted to the released data through directly fine-tuning on the released data.
>
> To minimize this risk, we plan to continuously update and expand our benchmark as a **live benchmark**. Specifically, we will employ the data collection pipeline described in Section 3.2 to gather and incorporate more data periodically, ensuring that future versions of the benchmark remain diverse and representative of real-world conditions.
>
> In addition, we intend to introduce **hidden test splits for evaluation only** in future releases to further prevent overfitting.
>
> We believe these measures will help maintain the benchmark’s long-term value and fairness for the research community.

---

> ### Author Response · Authors · 2025-11-25
> **Response to Question 5**
>
> We sincerely thank the reviewer for carefully reading the manuscript and pointing out these issues. We have thoroughly proofread the revised version and corrected all typographical, grammatical, and stylistic errors to improve overall readability and clarity.

---

### Meta-Review · Area_Chair_C9fZ · 2026-01-02

**Summary:**

This paper presents MIRAGE benchmark to evaluate the ability of LLM-based RAG systems to dynamically interleave multi-hop retrieval with evidence-grounded reasoning. With 579 tasks across 4 domains, MIRAGE simulates realistic, complex workflows where agents must iteratively generate sub-queries and navigate adversarial corpus to solve problems. While interleaved agents outperform single-turn baselines, there are still some headrooms for further improvements. Based on these points, the authors position MIRAGE as a critical resource for advancing robust agentic reasoning capabilities.

**Reviewer Concerns:**

Some reviewers have concerns on the number of samples in the benchmarks. While the authors responded as 4*500 = 2000 single-hop questions, 500+ trajectories are still not enough for the rigorous benchmarking. Some reviewers also have concerns on whether the benchmarks are different enough from the other benchmarks. The authors clarified the novelty over the other benchmarks and as an AC, those points make sense. Lastly, some reviewers raised the concerns on the difficulties of the proposed benchmark because the SOTA LLMs with agentic framework can already solve this benchmarks more than 60%. This can reduce the contributions of the proposed benchmarks.

**Reviewer Scores:**

The initial scores are 6,6,2. Unfortunately, the last reviewer (with score 2) does not reply to the author's response. Thus, AC is carefully taking into the response to that last reviewer. Unfortunately, AC agrees more on the reviewer side and the responses do not resolve the concerns that are raised by the last reviewer (e.g., # of samples, difficulty).

---

### Decision · Program_Chairs · 2026-01-26

Reject